# On the post-glacial spread of human commensal *Arabidopsis thaliana*

Cheng-Ruei Lee[1,2], Hannes Svardal[1], Ashley Farlow[1], Moises Exposito-Alonso[3], Wei Ding[3], Polina Novikova[1], Carlos Alonso-Blanco[4], Detlef Weigel[3] & Magnus Nordborg[1]

Recent work has shown that *Arabidopsis thaliana* contains genetic groups originating from different ice age refugia, with one particular group comprising over 95% of the current worldwide population. In Europe, relicts of other groups can be found in local populations along the Mediterranean Sea. Here we provide evidence that these 'relicts' occupied post-glacial Eurasia first and were later replaced by the invading 'non-relicts', which expanded through the east–west axis of Eurasia, leaving traces of admixture in the north and south of the species range. The non-relict expansion was likely associated with human activity and led to a demographic replacement similar to what occurred in humans. Introgressed genomic regions from relicts are associated with flowering time and enriched for genes associated with environmental conditions, such as root cap development or metal ion trans-membrane transport, which suggest that admixture with locally adapted relicts helped the non-relicts colonize new habitats.

[1] Gregor Mendel Institute, Austrian Academy of Sciences, Vienna Biocenter (VBC), Dr Bohr-Gasse 3, 1030 Vienna, Austria. [2] Institute of Ecology and Evolutionary Biology & Institute of Plant Biology, National Taiwan University, No. 1, Section 4, Roosevelt Rd, 10617 Taipei, Taiwan. [3] Max Planck Institute for Developmental Biology, Spemannstrasse 35, 72076 Tübingen, Germany. [4] Departamento de Genética Molecular de Plantas, Centro Nacional de Biotecnología (CNB), Consejo Superior de Investigaciones Científicas (CSIC), Madrid 28049, Spain. Correspondence and requests for materials should be addressed to C.R.L. (email: chengrueilee@ntu.edu.tw) or to M.N. (email: magnus.nordborg@gmi.oeaw.ac.at).

Large demographic turnovers can have enormous impact on species-wide polymorphism. The best-known example comes from humans: anatomically modern humans migrated from Africa and replaced existing forms such as Neanderthals and Denisovans, in what is effectively a 'genome-wide sweep'—the genomes of one population rapidly replace those of others. Note that the process differs from a classical genetic sweep not only in that it is effectively genome-wide, but also in that the driving force may not be genetics, but an environmental (or cultural) advantage: neutral variants can be swept to fixation not because they are linked to selectively advantageous alleles at particular loci, but because they are associated with an environment (which could be cultural) that has a large reproductive advantage. Human history has many examples of this. As long as there is some interbreeding, it is possible for native alleles to survive such an onslaught, either by chance, or because they are selectively favored (they may, for example, be locally adapted). Recent studies have found evidence of such residual genomic regions from Neanderthals or Denisovans in modern humans[1–7].

Here we study an analogous process in the model plant *Arabidopsis thaliana*, which is considered a weed and is associated with human disturbance throughout its range. Several studies have investigated its demographic history using molecular markers[8–14], and recently the genome-wide pattern of variation has been elucidated using genome-wide single nucleotide polymorphism (SNP) data[15–18]. The most comprehensive study to date, and the basis for the present paper, is re-sequencing the genomes of 1,135 worldwide accessions[18]. Simple comparison of genetic distances identified several accessions that are distantly related to the majority. These accessions were only found in the southern edge of the known species distribution and formed four geographically and genetically distinct groups (Cape Verde, Iberia, Sicily and Lebanon). *A. thaliana* thus comprises five equally distantly related genetic groups (in the samples used; further groups may well be discovered), one of which has a world-wide distribution and comprises over 95% of the current species. Because further analyses suggested that the five groups were the descendants of different ice age refugia, members of the four non-majority groups were referred to as 'relicts'. Interestingly, while most present-day *A. thaliana* are human commensals, relicts are associated with relatively undisturbed habitats, suggesting that they may be wild survivors of the expansion of humans and their commensals[19].

In this study we investigate this post-glacial expansion in greater detail. We first identify the traces of hybridization between relicts and non-relicts and investigate whether the pattern of introgression appears to have been associated with adaptation to the local environment. We then explore the geographic patterns of relict introgression, focusing in particular on finding traces of extinct relicts in non-relict genomes, and on inferring the possible origin of non-relict expansion. Our overall conclusion is that *A. thaliana* of today is the product of a dramatic series of range expansions, admixture, and local adaptation, and the current pattern of *A. thaliana* genomic variation is very different from that before the last ice age.

## Results

**Evidence of hybridization between relicts and non-relicts**. Of the 1,135 sequenced accessions[18] we used 1,002 from Eurasia and North Africa, the native range of *A. thaliana*. The rest, mostly from North America with a few from Japan, have almost certainly been introduced by humans in the last few hundred years[20,21]. The distribution of pairwise differences had two distinct peaks (Supplementary Fig. 1), as previously shown[18]. While the previous study identified five relict groups (Iberia, Cape Verde, Canary Island, Sicily and Lebanon), our analyses based on protein coding sequences (see 'Methods' section) showed the Canary Island and three Iberian accessions are probably of hybrid origin, resulting in four distinct relict groups.

We first focused on comparing relicts and non-relicts in Iberia given the large sample size of both groups in this region. We used French lines (which, like most worldwide *A. thaliana*, are non-relicts) as putative source for the Iberian non-relicts. ADMIXTURE with $K = 2$ confirmed the strong differentiation between relicts and non-relicts (as previously noted, Iberian non-relicts are more closely related to lines from Kazakhstan than they are to the geographically proximal Iberian relicts[18]), but also identified strong evidence of hybridization, with several Iberian non-relicts containing traces of relict genome (Fig. 1a). We performed population-based ABBA–BABA tests of the model (((French, Iberian non-relicts), Iberian relicts), *Arabidopsis lyrata*). If the population allele frequency differences do not deviate from the assumed tree relationship, the test-statistic, $D$, should be close to zero. A significantly positive ($Z \geq 3$) $D$ value indicates possible gene flow between Iberian relicts and non-relicts, while a significantly negative ($Z \leq -3$) $D$ indicates gene flow between Iberian relicts and French accessions. We found strong evidence of gene flow between relicts and non-relicts in Iberia ($D = 0.074$, $Z = 13.11$). Similarly, the three-population test ($Z \leq -3$ as significant)[22] supports Iberian non-relicts being a hybrid population between French non-relicts and Iberian relicts ($f_3 = -0.029$, $Z = -10.02$). To visualize the pattern of introgression across genome, we calculated the probability of originating from either parental population based on SNP allele frequency in each 10 kb window for each accession. Long tracts of relict introgression could clearly be observed in Iberian non-relicts, as well as some introgression in the other direction (Supplementary Fig. 2).

**Local adaptation resisted non-relict invasion into Iberia**. In humans, introgression from Denisovans appears to have contributed to high-altitude adaptation in modern Tibetans[7]. This observation can also be viewed as the resistance of locally adaptive Denisovan alleles to the migration pressure from modern humans. To look for analogous cases in Iberian non-relict *A. thaliana*, we performed genome-wide scans for SNPs associated with local climate across all Eurasian non-relicts: we used the second principal component axis (PC2) from altitude and 19 bioclim variables from WorldClim[23] because this PC axis separates the cold and wet northern from the hot and dry southern environments where all relicts currently locate. Because data for flowering time, an important adaptive trait, were available[24], we also searched for SNPs associated with this phenotype in Iberian non-relicts. All scans identified the same strong candidate (Supplementary Fig. 3a–c): a SNP (chr4:10999188) located < 2 kb from the flowering-time gene *TWIN SISTER OF FT* (*TSF*, AT4G20370) and 6 kb from a defence and drought associated gene *LESION SIMULATING DISEASE 1* (*LSD1*, AT4G20380), around which other studies have also identified GWAS peaks for plant phenology as well as local adaptation[25–27]. The non-reference allele is mostly found in the hot and dry Mediterranean environments (Fig. 1b), and its frequency is 0.92 in Iberian relicts, 0.44 in Iberian non-relicts, and 0 in French non-relicts, suggesting that this allele was introgressed from Iberian relicts into non-relicts. Among all 106,501 SNPs highly differentiated between Iberian relicts and French non-relicts, this SNP shows an unusually high

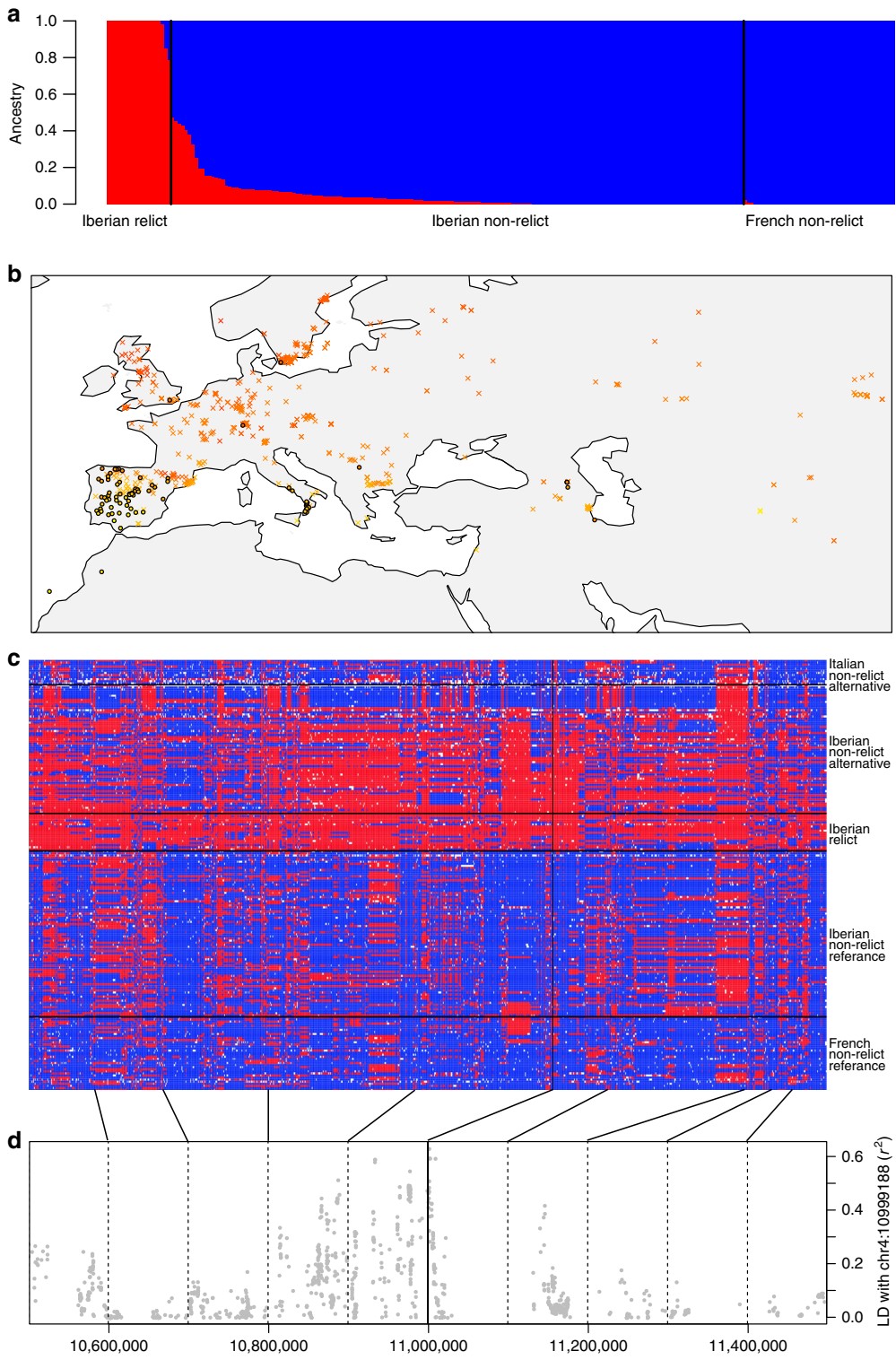

**Figure 1 | Admixture and adaptive introgression between relicts and non-relicts in Iberia.** (**a**) ADMIXTURE with $K = 2$ separates Iberian relicts and French non-relicts and shows the hybrid origin of Iberian non-relicts. (**b**) Allelic distribution of chr4:10999188, the SNP with highest climate and phenology GWAS scores. The colour gradient is based on climate PC2, separating southern hot and dry (yellow) versus northern cold and wet (red) environments. Two alleles are labelled as X (reference) or solid dots with black outline (alternative allele). (**c**) 1 Mb haplotype around this SNP (vertical black line). Columns are SNPs whose major allele is different between Iberian relicts and French non-relicts, and rows are accessions with black horizontal lines separating populations. The 'reference' or 'alternative' labels below each population denote the allele each population has in this SNP. Red and blue: major allele in Iberian relicts and French non-relicts respectively, white: missing. Iberian non-relicts with the alternative allele mostly contain haplotypes from Iberian relicts and some from Italy. (**d**) SNP LD with chr4:10999188 across the 1 Mb region. Solid vertical line is the location of chr4:10999188, and dashed vertical lines mark every 100 kb away from that SNP. The map was created with data from package 'rworldmap' of R.

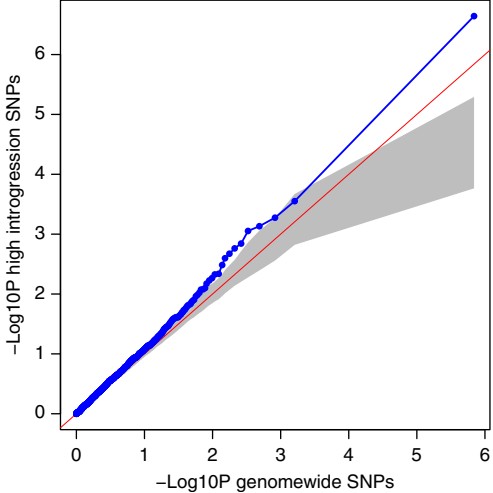

**Figure 2 | High-introgression SNPs show greater association with flowering time than background SNPs.** The QQ-plot shows the enrichment of high flowering-time GWAS scores of 'SNPs with high Iberian relict introgression into Iberian non-relicts' versus 'genomic SNPs with similar allele frequencies'. This analysis excludes the 1 Mb region near SNP chr4:10999188. Blue dots and line represent true value distribution, and the grey area denotes 5% significance thresholds based on 1,000 permutations.

degree of introgression (96th percentile). The introgression is clearly visible in the pattern of haplotype sharing around the SNP (Fig. 1c): while haplotypes of Iberian non-relicts with the reference allele resemble French lines, those with the alternative allele share haplotypes with local relicts. Some Iberian non-relict haplotypes with the alternative allele were similar to those in Italy, suggesting some introgression from other Mediterranean populations (Fig. 1c). Among Iberian non-relicts, there is extensive linkage disequilibrium (LD) in the 300 kb window around this SNP (Fig. 1d), as expected under admixture, and the average degree of introgression for differentiated SNPs within this window is very high compared with rest of the genome (98th percentile, see 'Methods'). These results are consistent with a scenario under which an allele adapted to local environments resisted the invasion of non-native genomes, and, as result, became introgressed into the invading gene pool, in a process analogous to what may have helped the high-altitude adapted allele reach high frequency in modern Tibetans.

Considering that the above example may be extreme in terms of effect size, and that many more subtle cases may exist, we examined the general relationship between introgression from relicts and flowering time GWAS scores in Iberian non-relicts. The pattern does seem to be more general: the top 25% SNPs in terms of introgression are much more likely to be associated with flowering time than are random SNPs with similar allele frequencies (see 'Methods'), even after excluding the 1 Mb region around chr4:10999188 (Fig. 2, Supplementary Fig. 3d). This suggests that selection also affected polymorphisms with weaker effect on flowering time.

To identify other traits that may have behaved similarly, we looked for gene ontology (GO) enrichments in windows also with top 25% highest relict introgression in Iberian non-relicts (Supplementary Fig. 2 and 'Methods'). Such windows were enriched for a number GO terms, in particular root cap development and trans-membrane transport (Table 1), suggesting that the Iberian relicts might have been better adapted than the early invaders with respect to traits other

than flowering, and that non-relicts currently inhabiting Iberia received the adaptive allele from relicts.

**Global patterns of relict admixture.** Both relicts and non-relicts co-exist in Iberia today, and this makes it relatively straightforward to identify introgressed regions. But what if introgression also happened in other parts of the world, and the local relicts went extinct? Could we still detect traces of introgression? This situation is similar to finding Neanderthal admixture in modern humans, except that there are no *A. thaliana* fossils we can extract ancient DNA from. The Mediterranean relicts were identified based on their average genome-wide genetic distance to other accessions (Supplementary Fig. 1). A simple approach is thus to apply the same logic and identify outlier accessions in short windows across the genome (we used 10 kb). From pairwise genetic distances between the 1,002 accessions in each window, we identified accessions that are very genetically distant from all others and called the corresponding haplotype an 'outlier haplotype'. If we count the number of outlier haplotypes identified in each accession, we might be able to identify admixed individuals and estimate the extent of relict introgression. This approach seems to work well: consistent results were obtained under a wide range of parameters (Supplementary Fig. 4), and the known admixed individuals on the Iberian Peninsula clearly stand out (Fig. 3a). Specifically, Iberian non-relict accessions with more outlier haplotypes tend to have higher extent of relict introgression as estimated from ADMIXTURE and ABBA–BABA *D* statistic (Fig. 3a).

Globally, non-relicts with putatively higher relict introgression were enriched in the north and south of the species distribution (Fig. 3b,c, regression of 10 kb outlier haplotype number on latitude, quadratic term $P < 0.001$)—even though all known relicts are located in the south. The pattern holds even after excluding relicts and Iberians (Fig. 3d, Supplementary Fig. 5a, $P < 0.001$) and suggests a scenario where non-relicts expanded mainly through the east–west axis across Eurasia and swamped local relict populations, whereas relicts in north and south had faced slightly lower immigration pressure, possibly due to climate differences. This is consistent with the notion that *A. thaliana* spread with agriculture[9] and more generally with the expansion of human commensals across Eurasia[19,28].

A few confounding factors could potentially affect our results: (1) given isolation-by-distance, we might identify two ends of the continuous genetic variation among non-relicts. However, the major axis of genomic variation in *A. thaliana* is east–west, not north–south[8,10,16,18], and we do not find more distant haplotypes in the eastern and western end of species distribution. Furthermore, if north–south isolation-by-distance contributed to these patterns, we would expect outlier haplotypes in the north to be highly divergent from those in the south because they are two extreme ends of a spatial continuum. As our later analyses showed, however, most northern outlier haplotypes are genetically close to those identified in the south (see below and Fig. 4). (2) Spatial sampling of the 1,002 accessions is highly uneven across Eurasia. We performed resampling analyses using thinned samples, controlling sample size in each 5° longitude-by-latitude geographical grid, and the results are consistent (Fig. 3d, Supplementary Fig. 5). (3) Populations in the periphery of species distribution might have smaller effective populations size (lower overall polymorphism $\pi_{ALL}$) and thus be unable to purge deleterious polymorphisms (higher non-synonymous to synonymous polymorphism ratio $\pi_N/\pi_S$). This excess of deleterious variation might cause peripheral accessions to be slightly different

**Table 1 | GO enrichment of genes in 10 kb windows with high relict introgression into the Iberian non-relict genomes.**

| Domain | GO term | Description | Fold enrichment |
|---|---|---|---|
| BIO | GO:0048829 | Root cap development | 8.28 |
| BIO | GO:1903506 | Regulation of nucleic acid-templated transcription | 5.31 |
| BIO | GO:0009686 | Gibberellin biosynthetic process | 3.35 |
| BIO | GO:0098655 | Cation transmembrane transport | 2.72 |
| BIO | GO:0000041 | Transition metal ion transport | 2.35 |
| BIO | GO:0009863 | Salicylic acid mediated signalling pathway | 2.19 |
| MOL | GO:0052381 | tRNA dimethylallyltransferase activity | 10.84 |
| MOL | GO:0015095 | Magnesium ion transmembrane transporter activity | 9.48 |
| MOL | GO:0008237 | Metallopeptidase activity | 7.16 |
| MOL | GO:0046873 | Metal ion transmembrane transporter activity | 5.29 |
| MOL | GO:0003713 | Transcription coactivator activity | 3.68 |
| MOL | GO:0015144 | Carbohydrate transmembrane transporter activity | 2.80 |

BIO, biological process; GO, gene ontology; MOL, molecular function.
Shown are significant GO terms from Fisher's exact tests with $P$ value ≤ 0.002 from 1,000 permutations, meaning that at most two permuted data have more extreme value than true data.

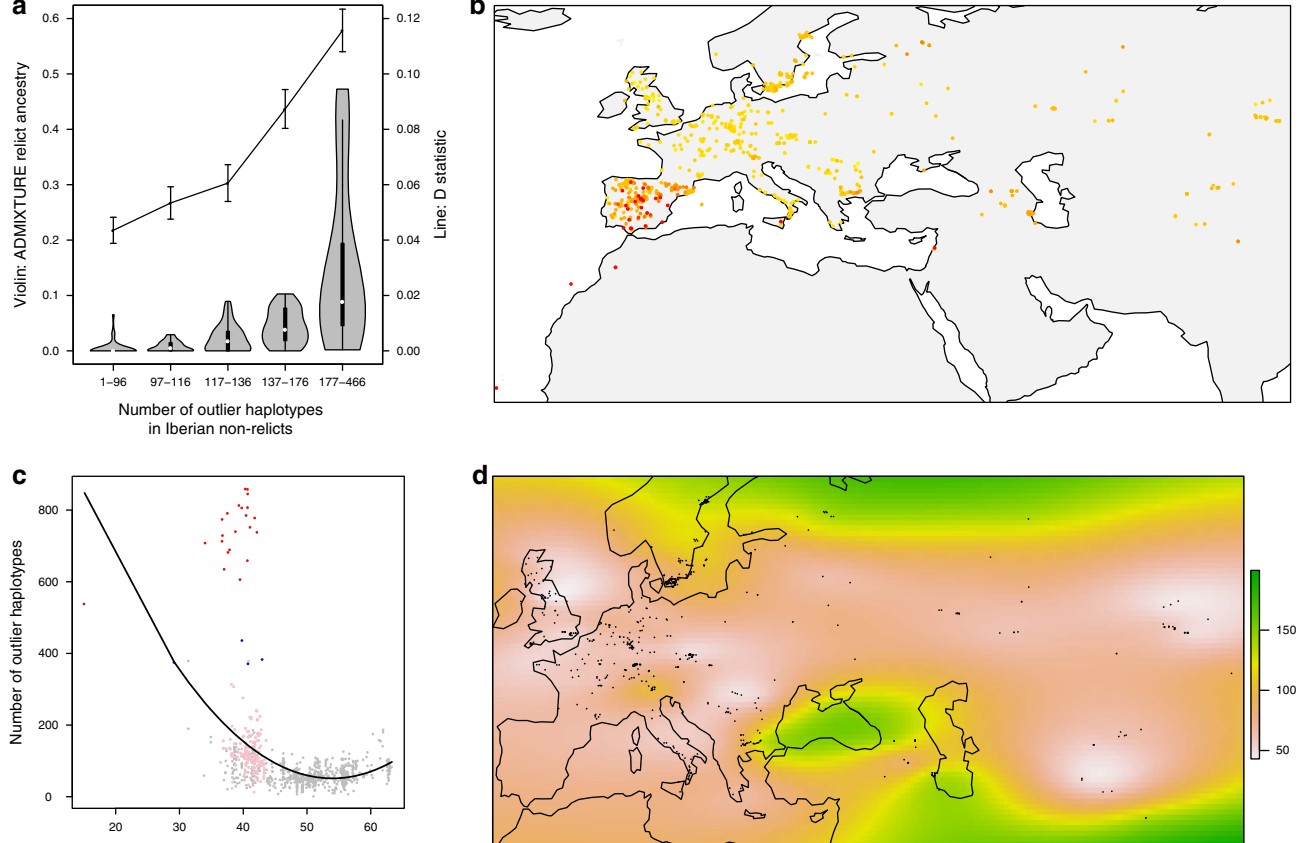

**Figure 3 | Global pattern of outlier relict haplotypes in *A. thaliana* genome.** (**a**) Iberian non-relicts were separated into five equally sized groups based on their 10 kb outlier haplotype numbers. Shown is the relationship between outlier haplotype numbers, proportional relict ancestry from ADMIXTURE, and $D$ statistics testing gene flow between Iberian relicts and non-relicts (error bar: s.d. from jackknife resampling). (**b**) Geographic distribution of accessions in the native range. Colour represents the number of outlier haplotypes in each accession, ranging from 0 (yellow) to ≥ 400 (red). (**c**) Accessions in the north and south of the species range tend to have higher numbers of outlier haplotypes. Red: Iberian relicts. Pink: Iberian non-relicts. Blue: Accessions defined as relict previously[18] but not in this study. (**d**) The density of outlier haplotypes across Eurasia, using thinned samples to take uneven sampling into account, and with relicts and Iberian non-relicts excluded due to their exceptionally high number of outliers (see 'Methods'). The maps were created with data from the package 'rworldmap' of R.

from others. However, among all non-relict populations, central Asia had the lowest $\pi_{ALL}$ and highest $\pi_N/\pi_S$, followed by southern Italy, and none of these show evidence of high relict introgression (Supplementary Fig. 6).

While the outlier analysis suggests admixture with relicts at both high and low latitudes, it tells us nothing about what those relicts may have looked like. Were they like one of the current relict populations, or did they belong to an unknown

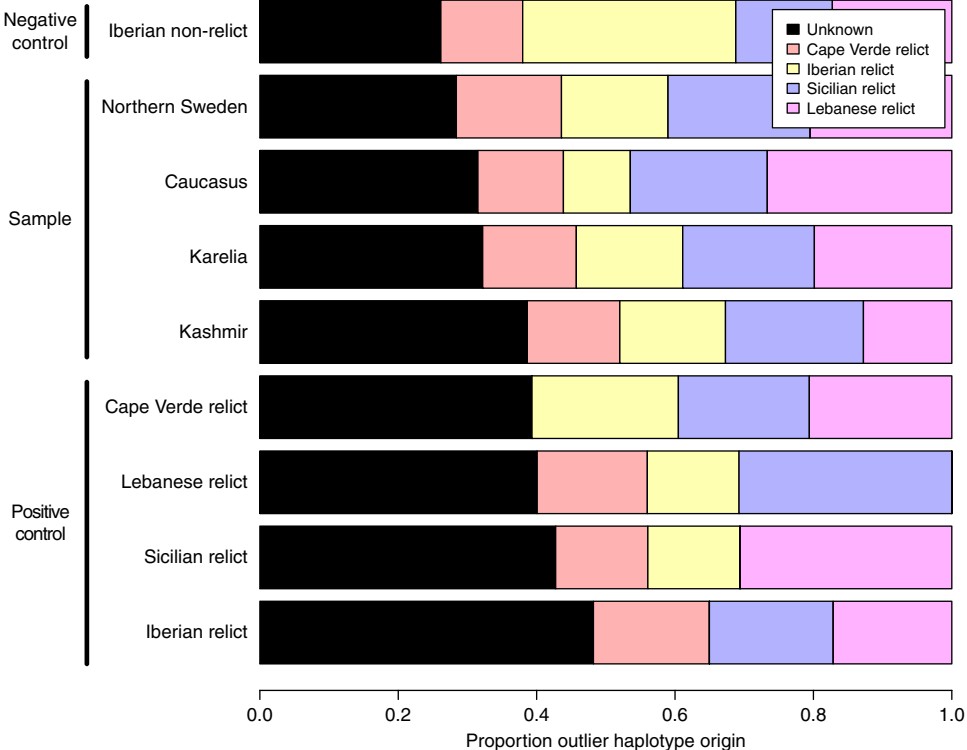

**Figure 4 | Possible origin of 10 kb outlier haplotypes.** The origin of each outlier haplotype in each local population (a horizontal bar) was assigned to one of the four known relict groups or an unknown relict. For each relict group (the four lower bar), the comparison was made only with the three other relict groups without to itself, serving as positive controls (origin of most outlier haplotypes unknown). Iberian non-relicts (the top bar) were used as negative control, where the origin of outlier haplotypes is mostly known. The only significant difference of proportion of unassigned outlier haplotypes (black) is between Karelia and Kashmir ($P < 0.001$, Fisher's exact test from random resampling 1,000 outlier haplotypes in each population).

group that no longer exists? This is a very difficult question to answer. The problem is similar to that of Vernot et al.[29] who first identified highly diverged haplotypes in modern humans and later assigned them either to Neanderthals or Denisovans, but with the additional complication that we also may need to assign outlier haplotypes to completely unknown groups. To accomplish this, we assigned each outlier haplotype to the known relict group it was genetically closest to, but left it unassigned (that is, assigned to 'unknown') if the distance to all known relict groups was greater than the distance used to define outliers in the first place (Supplementary Fig. 7). We then compared the fraction of unassigned haplotypes between different geographic regions (Fig. 4). We used the Iberian non-relicts as a negative control. In this population, the source of introgression is known (mainly Iberian relicts), and thus the fraction of unassigned outlier haplotypes (in the sense given above) serves as lower bound for what we might expect to see if there is no introgression from an unknown source.

At the other end of the spectrum, we use the known relict population as a form of 'positive control'. For each such population, we assign outlier haplotypes to the other three relict populations (that is, not allowing them to be assigned to their own populations—their true origin). The proportion of unassigned outliers after this procedure should give us an idea of what to expect when the true source of the outlier haplotypes is missing.

All populations were ranked by their 'proportion unassigned' in Fig. 4. We used Fisher's exact test to test for the difference in proportion unassigned between neighbouring populations, and the only significant result is between Karelia and Kashmir ($P < 0.001$). Thus, northern Sweden, the Caucasus and Karelia all have similar proportion of outlier haplotypes with unknown origin as the

negative control but much lower than the positive controls. The exception is Kashmir, which has a similarly high proportion unassigned as the positive controls. From this we conclude that the outlier haplotypes in northern Sweden, Karelia, and the Caucasus look no different from the outlier haplotypes present in the current relict populations, but that Kashmir is different. This is consistent with a scenario under which the relict populations expanded north after the last glacial maximum and the later non-relict expansion replaced relicts in mid-latitude, perhaps in association with the spread of agriculture or other human activities[9]. Supplementary Figure 8 shows examples from three 10 kb windows where the northern outlier haplotypes clearly had southern origins. The different results for Kashmir suggest that an unknown relict population may have contributed to the present-day polymorphism in this region.

**Finding the origin of the non-relicts**. We applied a previously established method to identify the origin of the non-relict expansion[9,30]. Briefly, assuming that there was a place of single origin, it should have higher genetic variation than the expansion front due to repeated founder events during the expansion. It is possible to search for such a centre of diversity systematically, as has previously been demonstrated in A. thaliana using a much smaller sample of non-relict accessions[9].

However, if one were to use this method directly on our data set, Iberia would be erroneously identified as the origin, because it contains an admixture zone between relicts and non-relicts and thus has very high diversity. To avoid this problem, we exclude relicts, accessions with high relict introgression, and all 10 kb outlier haplotypes before analysing the data. The results suggested the

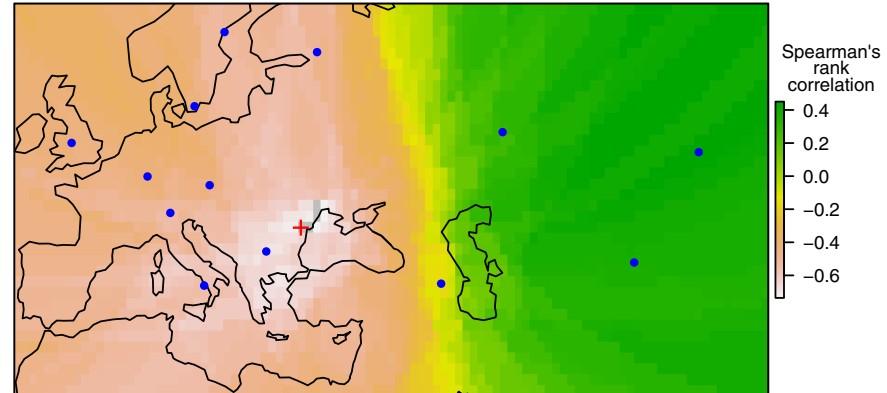

**Figure 5 | Origin of non-relict expansion based on whole-genome population polymorphism.** Grey denotes area with 1% lowest values, and the red cross is the most likely origin. Populations and accessions with high number of outlier haplotypes, as well as individual outlier haplotypes, were excluded to minimize relict influence. Blue dots represent non-relict populations. Colour scale represents Spearman's rank correlation between population polymorphisms $\pi$ and the geographic distances from all populations to each location. The map was created with data from package 'rworldmap' of R.

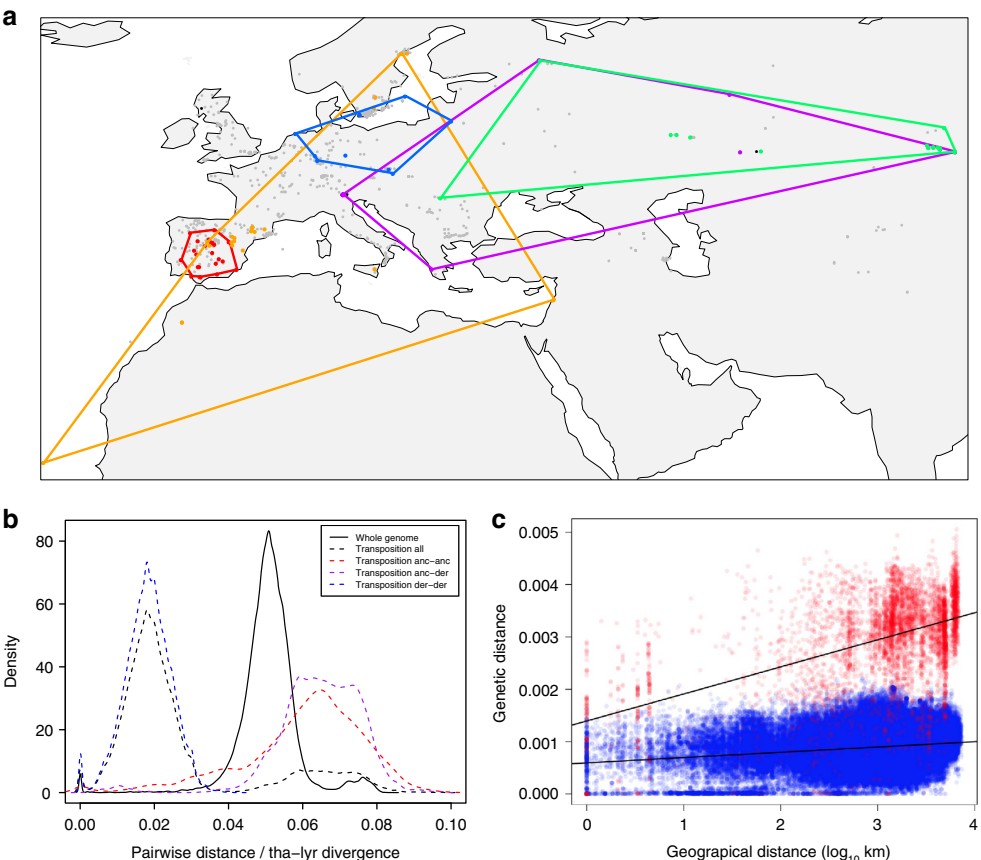

**Figure 6 | Genetic differentiation in the transposition region of chromosome 1.** (**a**) Geographical distribution of accessions with the derived (grey dots) or ancestral (coloured dots) haplotypes. The five different colours denote five ADMIXTURE groups among the latter. Black dots are admixed ancestral haplotypes. (**b**) Pairwise distance distribution for the whole genome (solid line) and the transposition region (dashed line), showing that differences among ancestral haplotypes are comparable to the whole-genome difference between relicts and non-relicts. (**c**) Ancestral haplotypes show stronger pattern of isolation by distance (red dots) than derived haplotypes do (blue dots). The map was created with data from package 'rworldmap' of R.

Balkans or the Black Sea area as the most likely origin for the non-relict expansion (Fig. 5), consistent with previous results[9].

**Supporting evidence from a chromosomal transposition.** The genome-wide data are nicely complemented by a large transposition on chromosome 1, which suppresses recombination between transposition alleles in a $\sim 700\,\text{kb}$ region[17]. The derived allele is present at very high frequency in *A. thaliana* and shows strong signs of having been selectively favored[17,31]. Among our 1,002 accessions, we identified 888 with the derived allele and 114 with the ancestral one. Interestingly, the ancestral alleles

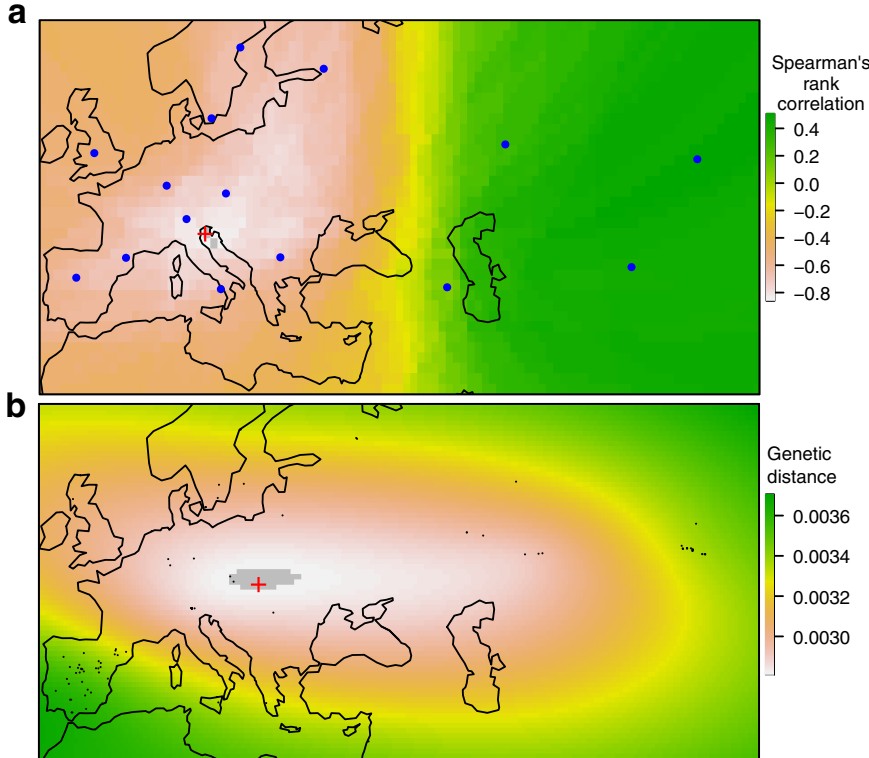

**Figure 7 | Origin of non-relict expansion based on the transposition in chromosome 1.** Grey denotes regions with 1% lowest value, and the red cross is the most likely origin. (**a**) Estimate based on population polymorphism of the derived haplotypes. Blue dots represent populations. Colour scale represents Spearman's rank correlation between population polymorphisms $\pi$ and the geographic distances from all populations to each location. (**b**) Estimate based on the genetic distances. Black dots represent accessions with the ancestral haplotype. Colour scale represents the genetic distance between local ancestral and derived haplotypes. The maps were created with data from package 'rworldmap' of R.

tended to be found in the southern and northern end of the species distribution (Fig. 6a), a pattern reminiscent of the outlier haplotype analysis (Fig. 3). In fact, all relict accessions carried the ancestral allele, and accessions with the ancestral allele had significantly more genome-wide outlier haplotypes than those that did not (mean = 234.6 and 73.0 respectively, ANOVA $F_{1,1,000} = 288.19$, $P < 0.001$), even after excluding relicts (mean = 113.3 and 73.0 respectively, ANOVA $F_{1,978} = 66.38$, $P < 0.001$). The derived transposition allele is therefore a unique feature of the non-relict genome and may have reached high species-wide frequency simply by being part of the non-relict expansion (although there is also strong evidence for selection directly on the transposition[17,32]). The estimated age of the transposition event is at least 43,000 years[17], consistent with it becoming fixed among the non-relicts during the last ice age. Thus, the ancestral haplotypes should reflect population structure before non-relict expansion, while the derived haplotypes may provide insight into the non-relict expansion.

Consistent with this, the level of polymorphism among derived haplotypes in the transposition region on chromosome 1 is very low, while the divergence between ancestral and derived haplotypes (or between different ancestral groups, see below) is as high as the genome-wide distance between relicts and non-relicts (Fig. 6b). In addition, much stronger isolation by distance exists among ancestral than among derived haplotypes (Fig. 6c), reminiscent of the whole-genome pattern[18].

ADMIXTURE[33] analysis clustered ancestral haplotypes in five haplogroups (Supplementary Fig. 9). As shown in Fig. 6a, this chromosomal region in all 19 Iberian relicts belonged to an Iberia-specific haplogroup (red dots in Fig. 6a). All other relicts (Cape Verde, Sicily and Lebanon) belonged to a haplogroup (orange in Fig. 6a) that also contained non-relict accessions from the Mediterranean region and Sweden. The presence of a haplogroup only in southern and northern, but not mid-latitude Europe, supports our conclusion that relicts (carrying this haplogroup) once occupied all of Europe before being replaced by non-relicts (carrying the chromosomal translocation, grey dots in Fig. 6a) that primarily spread longitudinally. The presence of some Iberian non-relicts in this orange group is also consistent with our observation that some Iberian non-relicts appear to carry haplotypes otherwise found in southern Italy (Fig. 1b). ADMIXTURE results with $K = 6$ or 7 show similar trends that northern haplotypes mostly have southern origins (Supplementary Fig. 10).

Turning to the derived allele, we used two methods to infer its origin. The first was the one used for genome-wide data above. The second tried to identify the ancestral haplotype most closely related to the derived ones, and used its location as an estimate for the origin of the derived alleles. Both methods yielded estimates consistent with the genome-wide estimate: the derived allele (Fig. 7), and the non-relicts as whole (Fig. 5), appears to have spread from the northern Balkans or central Europe.

## Discussion

The genomic effects of population expansion are best documented in humans. The expansion of anatomically modern humans out of Africa did not occur in a vacuum, because earlier groups, like Neanderthals and Denisovans, were already there. Although these groups were soon replaced by modern humans, there is strong evidence of hybridization, and introgressed genomic

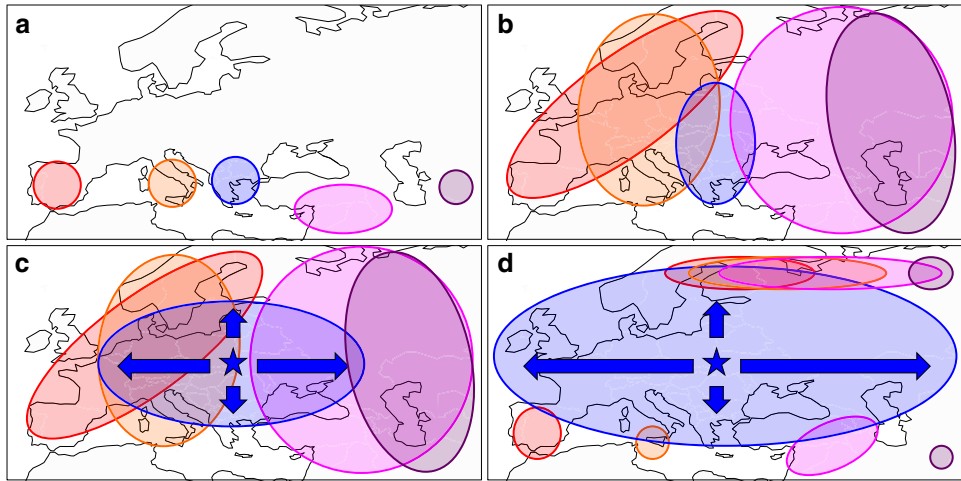

**Figure 8 | Possible demographic history creating the present-day pattern of spatial genetic variation in *Arabidopsis thaliana*.** (**a**) During the last glaciation, populations in different refugia diverged into separate groups. (**b**) After ice age, each group expanded northward. (**c**) Some time later, one group from northern Balkan and eastern Europe (the non-relicts) quickly expanded east–west, generating the present-day pattern that (**d**) relict genomic regions are mainly found in the south and north of species range. The maps were created with data from the package 'rworldmap' of R.

regions have been associated with important traits[1–7]. Farmers later replaced hunter-gatherers worldwide, continuing the process of population replacement[34]. Our history is full of large population migration and replacement events, the most notable example in recent time being the mass migration of Europeans into America and Australia over the past six centuries. In many cases, natives and, presumably, leftovers of their genomes were restricted to regions away from the main expansion wave. These events not only shaped the genomic patterns of variation in humans but also resulted in massive anthropogenic change of local landscapes, ecosystem, and the accompanying dispersal of domesticated species and other commensals[19,28].

Here we investigated the complex dynamics affecting spatial patterns of genomic variation in *A. thaliana*, a wild species that clearly benefits from human-induced disturbance. Our analyses suggest an evolutionary scenario that is illustrated in Fig. 8. In summary, populations from different glacial refugia appear to have expanded northwards (Fig. 8a,b) at the end of last ice age. This is supported by the results that most northern outlier haplotypes likely have southern origins, and in Sweden, the ancestral haplotypes of the chromosome 1 translocation belong to the same genetic group as Mediterranean relicts (Figs 4 and 6a). After the first wave of expansion reached what is today northern Sweden and Russia, a second wave started from a population (the ancestors of today's non-relicts) near northern Balkan and eastern Europe and expanded mainly along the east–west axis, erasing the trace of those first colonizers (different relict groups) in mid-latitudes (Fig. 8c), while leaving chunks of relict genomes in the south and north of the species range (Fig. 8d). Our results also suggested some leftover relict genomic chunks may confer local adaptation, at least in Iberia, where genomic regions with high relict introgression associate with flowering time and GO terms such as root cap development or iron transmembrane transport. It may thus well be the case that certain alleles that are adapted to local abiotic factors like climate or soil are able to resist such migration.

We emphasize that, although we have done our best to rule out alternative models, the model space in these types of historical analyses is infinite, and it is therefore impossible to 'prove' that the scenario in Fig. 8 is the right one. It is a model that fits the current data well (and we welcome attempts to find better ones—in particular as more samples from under-sampled regions become available).

Under our model, one would expect considerable divergence between the western and eastern non-relict expansion fronts, resulting in higher allele frequency differences between the east and west (so-called genetic surfing[35–37]). This is indeed observed: population structure separates east from west, with a boundary around eastern Europe[8,10,14,16,18]. Although this pattern has led others to conclude that present-day *A. thaliana* originated from refugial populations in central Asia and Iberia followed by secondary contact in eastern Europe, the facts that central Asia has least genetic variation, least unique alleles, slowest LD decay, and highest $\pi_N/\pi_S$ all suggest it having been colonized recently (Supplementary Fig. 6)[15,16,18].

It is tempting to speculate that human activities, especially agriculture, created environments favoring expansion of non-relicts and replacement of relicts. First, the date and origin of non-relict expansion correspond to the spread of agriculture in Europe (this study and ref. 9). Second, while non-relicts are human commensal weeds associated with disturbed habitats such as roadsides or farms, relicts inhabit relatively un-disturbed environments[18]. It is possible that, when agriculture replaced hunting and gathering, the changed landscape expanded suitable environments for non-relicts, thus altering the species-wide patterns of genetic variation in *A. thaliana*. Similarly, European settlers in America, Australia, and New Zealand have been cutting down native forests for agriculture, and their cattle have been grazing native grasslands, making room for Eurasia-originated weedy pioneer plants and effectively 'Europeanizing' fauna and flora in these places[19]. Therefore even in recent times humans are still 'helping' the worldwide expansion of non-relicts, as *A. thaliana* accessions in North America arrived only a few centuries ago and are genetically similar to non-relicts in northwestern Europe[18,20]. We do not expect the dynamics to stop. Just as the spatial pattern of genomic variation in *A. thaliana* was likely affected by human activities in the past[9,18], one expects a major disturbance of the current pattern by ongoing anthropogenic climate change.

## Methods

**Samples and genomic distances.** Genomes of 1,002 accessions from the native range of *A. thaliana*[18] were used. Unless otherwise stated, we focused the analyses on the coding regions of 20,186 genes where more than 80% of the gene length was sequenced in more than 80% of accessions. Sites with indels, more than two alleles,

or more than 10% missing data were further excluded. SNPs were classified as synonymous or non-synonymous by Variant Annotation Tool v.2.0.1 (ref. 38). Relicts were identified with the same method as before[18]: the median distance of each accession to all others was calculated, and relicts were defined as those who have exceptional high median distance to others.

**Hybridization between relicts and non-relicts in Iberia.** To test whether Iberian non-relicts ($n = 170$) were hybrids between Iberian relicts ($n = 19$) and French non-relicts ($n = 45$), we used ADMIXTURE[33] with $K = 2$ on these accessions. We performed 10 independent runs with different random number seeds and reported the one with lowest cross validation error. Note that we did not intend to estimate the best $K$ value and the accessions' corresponding population assignments. Rather, given the high divergence between relicts and non-relicts, unsupervised ADMIXTURE $K = 2$ would separate Iberian relicts from French non-relicts and estimate the proportional ancestry of Iberian non-relicts from either parental populations.

We further used the population-based version of ABBA–BABA test ($D$ statistics)[1,22,39,40] to test deviation from this population tree topology: (((French non-relicts, Iberian non-relicts), Iberian relicts), A. lyrata). This method is based on allele frequencies of the four populations instead of allele states of four individuals in standard $D$ statistics. Since the outgroup A. lyrata had only one sample, the reference strain MN47 (ref. 41), its allele frequency was assumed to be either 1 or 0. Significance was determined based on $Z$ scores using the block-jackknife method[1] with block size of 1 Mb. A significant positive value suggests introgression between Iberian relicts and non-relicts, and negative value suggests introgression between Iberian relicts and French non-relicts. In addition, we employed the three-population test[22] in the form of $f_3$ (Iberian non-relict; Iberian relict, French non-relict) in ADMIXTOOLS[42] to test for admixture. A significant negative value suggests Iberian non-relicts being hybrids from the two other populations. For both ABBA–BABA and the three-population test, results are regarded as significant if the absolute value of $Z$ scores is higher than three.

To visualize the extent and size of introgression, for each 10 kb window we estimated the probability that each individual descended from either parental population. A SNP was regarded informative if the major allele differed between Iberian relicts and French non-relicts. For each informative SNP in each individual, we calculated the probability of the individual's allele descending from the relict population as $p_r*(p_r + p_n)^{-1}$, where $p_r$ and $p_n$ are the frequencies of the target individual's allele in the relict and non-relict populations respectively. This value was then averaged across all informative SNPs within each window. Using the multiplication rather than the average of conditional probabilities produced similar results. Windows with < 5 informative SNPs were excluded.

**Relict introgression and adaptation in Iberia.** Based on the wide-held idea that SNPs strongly associated with climates were more likely to contribute to local adaptation[43,44], we first performed GWAS on climates of all Eurasian non-relict accessions. Altitude and 19 bioclim variables were retrieved from the WorldClim database[23], and principal component analysis (PCA) was performed on all 20 variables. We focused on the second principal component (PC2) because it separated the cold and wet northern versus the hot and dry southern Eurasia, representing a good environmental proxy for local adaptation in the Mediterranean region where relicts currently located. We further retrieved plant phenology traits from a previous study[24] and performed GWAS on 115 Iberian non-relicts whose traits were measured in that study. GWAS was done with MTMM[45], using a kinship matrix in a mixed model framework to correct for population structure.

Since the GWAS was performed with all bi-allelic SNPs in the genome, for the following analysis we also used all SNPs (instead of SNPs from the coding region of 20,186 genes) to facilitate the comparison between SNPs with high GWAS scores and those with high relict introgression. To identify SNPs with high relict introgression, we used a more stringent definition: a SNP was regarded as diagnostic between Iberian relicts and French non-relicts if the allele frequency is $\leq 0.25$ in one and $\geq 0.75$ in the other population. For each diagnostic SNP, we calculated the probability of relict introgression into Iberian non-relicts (the ancestry index) as: $(p_{IbeN} - p_{FraN})*(p_{IbeR} - p_{FraN})^{-1}$, where $p_{IbeN}$ was the allele frequency in Iberian non-relicts, $p_{FraN}$ in French non-relicts, and $p_{IbeR}$ in Iberian relicts. Those top 25% diagnostic SNPs with highest ancestry index were denoted as 'high-introgression SNPs'. Since the ancestry index and GWAS power both depended on allele frequency, we did not compare those high-introgression SNPs to all others in the genome. Instead, we identified SNPs whose allele frequencies were within the 95% range of allele frequency distribution of high-introgression SNPs as 'background SNPs'. With Q–Q plots, we compared the distribution of GWAS scores ($-\log_{10} P$ values) between high-introgression and background SNPs, and significance was determined by 1,000 permutations. In each permutation we ordered SNPs by their physical location and shifted the 'high-introgression or background' status vector by a random number down the genome, while keeping GWAS scores unshifted. This method preserved the relative order of SNPs and can more effectively control for the non-independence among neighbouring SNPs caused by LD or local SNP density. Note that in Fig. 2 the grey significance threshold slightly bends downwards, and we suspect this may be caused by the non-independence among SNPs, although we did not research this further.

To compare the magnitude of relict introgression into Iberian non-relicts in the 300 kb high-LD region around SNP chr4:10999188 with the rest of the genome, we calculated the average ancestry index of all diagnostic SNPs inside the 300 kb region and compare the observed value to those calculated from 1,000 randomly drawn 300 kb window from the genome.

We used similar logic to estimate the ancestry index for 10 kb windows, using SNPs from coding regions of the 20,186 genes. In the previous section, for each 10 kb window in each accession we calculated the probability that its haplotype descended from Iberian relicts (as opposed to from French non-relicts). Here in each window the probability was averaged among accessions within each of the three populations, and a diagnostic window was defined as one with mean probability $\geq 0.75$ in Iberian relicts and $\leq 0.25$ in French non-relicts. Ancestry index was calculated with the same equation above, and therefore the top 25% windows with highest ancestry index represented ones where haplotypes from Iberian relicts and French non-relicts were highly differentiated and where Iberian non-relict haplotypes were mostly descended from the relicts. We then compared the enrichment of GO terms between these high-introgression windows versus background ones. For each GO term, significance was estimated by Fisher's exact test with the same permutation method described above.

**Identifying genome-wide patterns of relict introgression.** To identify species-wide relict introgression, we used the original logic where relicts were identified: relicts were defined as those with high genomic distances to all others[18]. For every 10 kb window, the same procedure was applied as in the whole-genome data set: the median distance of each accession to all others was calculated, and among the distribution of 1,002 median values, an accession was defined as having a relict haplotype (an outlier haplotype) if it was more than one, two, or three s.d. away from the mean and if its median distance to all others exceeded 7% of average *thaliana–lyrata* divergence in that window. The three different criteria of choosing outliers give similar results (Supplementary Fig. 4), and we used the three standard deviation criterion for following analysis. We used 7% because all genome-wide distances between relicts and non-relicts exceeded 7% of average *thaliana–lyrata* divergence, and setting this minimum required divergence reduces the chance of identifying outlier windows caused by isolation by distance within non-relicts. This method was based on a reasonable assumption that the majority of the genome and accessions are non-relict[18], and haplotypes of relict origin behaved as outliers. This method did not require the relict parental information as the Iberia–France comparison and may identify residual relict haplotypes from any unknown relict group that did not exist among our samples. Note that a genomic region with highly diverged haplotypes could be caused by either long-term balancing selection or recent relict introgression. The former, however, should have much shorter haplotype length than the latter due to generations of recombination and was less likely to be identified by our 10 kb window size.

To address whether our identification of outlier haplotypes was affected by uneven spatial sampling of accessions, we re-analysed the data with 1,000 spatial resampling trials. Only 996 Eurasian accessions whose coordinates between latitude 35–65° N and longitude 10° W–90° E were analysed, and within the geographic region, the map was divided into 5° latitude by longitude grids. In each resampling trial, if a grid had more than 10 accessions, 10 were randomly sampled without replacement, otherwise all accessions in this grid were used (Supplementary Fig. 5c). The whole outlier haplotype analysis was applied on this resampled data set, and we compared accessions' number of outlier haplotypes with latitudinal origin among the 1,000 resampling trials. To summarize the results, for each accession we averaged the numbers of outlier haplotypes identified across the resampling trials it appeared. Using different grid sizes did not change our conclusion.

In addition, we used a grid-based approach. Each geographic grid, instead of each accession, was regarded as a unit in the outlier haplotype analysis. Relicts were first excluded, and grids with less than two accessions were also excluded. In each 10 kb window for a given pair of grids, the pairwise distances of accessions between grids were computed, and the average was taken as the 'genetic distance' between grids. If one regarded a grid as a 'population', this estimate is equivalent to $\pi_{XY}$ (ref. 46). Following the accession-based procedures, the outlier grids were identified in each 10 kb window, and we plotted number of outlier windows of each grid on the map.

**Origin of outlier haplotypes.** We performed this analysis to test whether outlier haplotypes within non-relicts were introgressed from the four previously defined relict groups (Cape Verde, Iberia, Sicily and Lebanon) or other unknown relicts not present in our samples, with emphasis on regions where outlier haplotypes were enriched (northern Sweden, Karelia, Caucasus and Kashmir). For every outlier haplotype in an accession, we estimated its average genetic distance to the four known relict groups and assigned the relict group with closest genetic distance as its origin. We did not include non-relicts as a potential origin because the haplotype was already identified as outlier from all non-relicts. If the genetic distance to the closest relict is still higher than the threshold used to define it as an outlier (see the previous section), the origin is set to unknown. For each population, the origin was estimated from all outlier haplotypes in all accessions, and we

compared the proportion of unknown origin among populations to infer possible introgression from an unknown relict.

Since previous results showed strong evidences of introgression from Iberian relicts (and possibly other Mediterranean relicts) into Iberian non-relicts, we estimated the origin of outlier haplotypes in Iberian non-relicts as a negative control, representing the proportion of unknown origin if the origin of all outlier haplotypes were known. The four relicts groups were also analysed while respectively excluding themselves from the reference, thus serving as positive controls when a significant proportion of outlier haplotypes had an unknown origin. If all outlier haplotypes in a population were descended from the four known relicts, the proportion of unknown origin should be significantly different from positive controls but not from negative controls, and vice versa.

We ranked the negative control, four positive controls, and the four target populations based on their proportion of unknown origin and tested for significant difference of proportional unknowns between neighbouring populations. Since the total number of outlier haplotypes varied among populations, a naïve test might tend to assign higher significance to comparisons with larger sample size. We therefore randomly resampled 1,000 outlier haplotypes with replacement from each population and use Fisher's exact test on this data set where all populations had the same sample size.

**Origin of the non-relict expansion.** In a simple population expansion model with one source of origin, populations closer to the expansion origin would tend to have higher genetic diversity. One could therefore investigate the relationship between a specific geographic location and all sampled populations. If this location was closer to the expansion origin, the correlation between polymorphism in sampled populations and their geographic distance to this location would be more negative[9,30], and the opposite is true if this specific geographic location is far from the origin.

Populations were defined from geographic regions[18]. To minimize the influence of relict introgression, we excluded all accessions from Iberia and Africa. For other individuals, we also exclude parts of their genome showing evidences of relict introgression: In an accession, if a 10 kb window was identified as a outlier haplotype from the previous analyses, all sites in this region of this accession were treated as missing. Since the relict introgression only constituted a small portion of non-relict genomes, considering these regions as missing effectively removed the interference from relict introgression without heavily biasing the estimate of pairwise genomic distances. Mean pairwise distances $\pi$ within each population were calculated from this filtered data.

For every geographic location between latitude 30–65° N and longitude 10° W–90° E, in a grid size of 1°, we calculated the Spearman's rank correlation between population polymorphisms $\pi$ and the geographic distances from all populations to this location. Geographic distance was calculated with function 'rdist.earth' in the 'fields' package of R (ref. 47). The region with the most negative correlation was regarded as the origin of non-relict expansion.

**The transposition on chromosome 1.** We investigated a chromosomal transposition between 20.27 and 21.03 Mb in chromosome 1 (refs 17,31) as a supporting evidence of our whole-genome analysis. Based on the exact transposition breakpoints identified previously[17], we examined the distribution of this transposition in the 1,002 accessions. Accessions were determined to have the ancestral or derived arrangement based on: (1) Illumina paired-end reads spanning breakpoints of the ancestral or derived haplotypes, (2) accessions' position in the neighbor-joining tree and (3) pairwise distances between accessions. All three methods had perfect agreement.

We used ADMIXTURE[33] to investigate the population structure of ancestral haplotypes. For each of $K$ values ranging from 2 to 15, we performed 10 independent runs with different random number seeds. We plotted the distribution of cross-validation errors of all $K$ values and performed simple $t$-tests of cross-validation errors between neighbouring $K$ values. The cross-validation errors continued decreasing with higher $K$ values, as expected for a self-fertilizing species with high population structure (Supplementary Fig. 9a). We chose $K = 5$ for our analyses because its cross-validation errors: (1) are relative low, (2) do not overlap much with the range of the previous $K$ and (3) have highly significant difference compared with the previous $K$ but not so much with the next $K$ value (Supplementary Fig. 9a). Among the 10 independent runs of $K = 5$, we reported accessions' population assignment from the run with lowest cross-validation error. An accession was assigned admixed (in this transposition region) if its proportional ancestry was below 0.5 in all populations. The ADMIXTURE assignment was also compared with results from principal coordinate analysis of pairwise genetic distances of the ancestral haplotypes.

We used two methods to investigate the geographical origin of derived haplotypes. The first method was equivalent to the whole-genome method before, this time using all derived haplotypes. The second method was based on the simple idea that, ancestral haplotypes from the geographical origin of derived ones should share a more recent common ancestor with derived haplotypes than other ancestral haplotypes do. The geographical origin of derived haplotypes should therefore have local ancestral haplotypes with closest genetic distance to all derived ones. We therefore calculated each ancestral haplotype's mean genetic distance to all derived ones and interpolated that across the Eurasia map using the thin

plate spline method implemented in the function 'Tps' of package 'fields' in R (ref. 47).

**Maps.** Figures using maps were created using 'rworldmap' in R (ref. 47).

**Data availability.** All data are available at http://1001genomes.org/ or available from the authors upon request.

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

## Acknowledgements

Supported by the ERC (MAXMAP, M.N.; IMMUNEMESIS, D.W.), a collaborative grant from Austrian Science Fund and DFG (SPP ADAPTOMICS; M.N., D.W.), Max Planck Society (D.W.), Austrian Academy of Sciences (M.N.), BIO2013-45407-P from the Ministerio de Economia y Competitividad of Spain (C.A.-B.), EMBO Long-Term Fellowship (C.-R.L.), and 105-2311-B-002-040-MY2 from Ministry of Science and Technology of Taiwan (C.-R.L.).

## Author contributions

C.-R.L. and M.N. conceived the study and wrote the manuscript. C.-R.L. performed the analyses with contributions from all authors.

## Additional information

**Competing financial interests:** The authors declare no competing financial interests.

