## [Peer Review File · Nature Communications]

Reviewers' Comments:

Reviewer #1 (Remarks to the Author)

The authors have presented a study of patterns of genomic variation and introgression among relict and non-relict populations of *Arabidopsis thaliana*. The subject matter and findings are interesting, and the authors have used some creative methods to study these patterns. But I find that the interpretation seems to go beyond what the data warrants in several important areas, as I describe below in "Major comments" #1 & #2. I like the study of the large effect locus that is highly differentiated with high LD and the description of how this region is introgressed, but there are some issues with respect to the broader use of GWAS that made it difficult to interpret the results (#5, below). The supporting evidence from the chromosomal transposition is quite interesting and it's nice to see that it supports some of the other patterns of introgression. In light of these comments, I think the manuscript requires extensive re-framing, but could be suitable for Nature Communications following such revision. It seems that many of the findings, however, are mainly confirming and extending previous results, so the novelty may not be significant enough for this journal (e.g. references 23-25 previously studied the major effect locus; relict vs. non-relict populations and human-associated spread studied in references 10, 19, 26; identification of the transposition in references 18,29).

Major comments:

1. The interpretation seems too ready to assume selection favouring the outlier regions from the relicts without exploring alternative possibilities. For example, presumably, recombination rate affects the degree of introgression or at least the blockiness of introgression, and there are hotspots and coldspots in the genome. It therefore seems possible that the GO term analysis identified genes and GO-terms that co-vary with cold spots or hot-spots? Jumping to adaptive explanations without considering alternatives seems premature. Similarly, I think the outlier-based trends (GWAS + introgression) could plausibly be explained by pure demography rather than introgression + selection, and more care needs to be taken to recognize this possibility (see #5, below).
2. Assuming the introgression hypothesis is true, what evidence actually pegs the expansion of non-relicts to the original expansion into Europe? There have been many waves of human movement back and forth over the ages, and it seems just as likely that an increase in trade during, say, the middle ages or the industrial revolution, was responsible for the observed introgression patterns (or more likely, very complex patterns of human-associated movement). The authors seem inordinately eager to link this with the *Homo sapiens* vs. Neanderthal introgression story (e.g. line 170-173) and post-glacial expansion, when there is really only conjecture to support the case one way or the other. I think a lot of reframing and toning down in the Discussion is necessary to be more scientifically rigorous here. In particular, I find the paragraph from lines 336-353 is WAY too speculative. For example, saying that the "northward expansion certainly happened in Sicily and Lebanon relict" is putting too much certainty in these results. This statement arises from the result that $K = 5$ and is based on genotyped accessions, but other relatives that were not included in this panel could have been the source of such northern populations, and if $K \neq 5$ other patterns might be seen. Overall, the paper explores one hypothesis that is consistent with the data, but this does not mean that it is the best explanation, as many other possible hypotheses have not been tested and much data is not included here. More caveats are necessary with observational studies such as this.
3. While the explanation for the pattern observed in Figure 3C seems consistent with the data (line 186-195), other explanations might also make sense. The fundamental pattern here is that when

you identify regions of the genome that are outliers in terms of genetic distance, then count the number of outliers, you see more outliers in extreme northern and southern regions. As noted on line 200, the same pattern could be seen with isolation-by-distance, but the authors discount this because the geographic distance is longer on the east-west axis than the north-south axis. I do not find this discounting entirely convincing, because presumably as a human associated plant, it has been dispersing with human migration, which may have gone much more often east-west than north-south. Geographic distance is less appropriate than "human migration distance" (or something like that). As I mentioned above, human migration is complicated and the simplest story may not be the most accurate one. I find that there is a bit of "just-so story" here and lack of testing against other hypotheses, and I would like to see much more hedging.

4. I am not a fan of the "methods last" style of journal article that has become more popular in the past 10 or so years, but this is hardly the authors' fault. However, I think it would be helpful to have at least one paragraph summarizing the approaches that are being used, perhaps at the end of the introduction. The results section feels like a very abrupt transition. In some cases there are results that are introduced with little context for how the method work. For for example, what are "outlier haplotypes" (line 179)? I can guess, but going to the methods should only be necessary to understand the details of the method, not to understand roughly how it is working. Please try to put more context for the methods in the actual paper, as I don't think page counts are very restrictive for this journal.

5. How was the GWAS done? The methods are very sparse. What program was used? Was there a correction for population structure? If not, some consideration must be made for how co-variation between environment and underlying population structure may bias the results. The lack of detail here made it difficult to assess some of the findings. If population structure covarying with environment is driving the GWAS patterns, then presumably this could also be driving many of the patterns of covariation with blocks of extreme introgression. I think this is unlikely to explain the particularly strong single locus effect, but could be much more important for the other results. For example, on lines 150-157, "the top 25% of SNPs in terms of introgression are more likely to be associated with flowering time than are random SNPs". Assuming there has been no correction for population structure in the GWAS, couldn't this be driven by drift? The phenotype varies along a spatial gradient, as does introgression. Therefore, spurious associations between neutral alleles and phenotypes could occur just due to drift, and these would also be likely to be outliers in terms of introgression. GWAS without correction for population structure will just identify those loci that are most strongly structured in such case, so of course the same ones would be found in each analysis. I couldn't see any way that the analysis had accounted for this very important issue.

Minor comments:

- line 22: the genome is a product of a dynamic equilibrium, but is not itself an equilibrium. It is a physical thing.

- this use of "genome" seems strange, as it represents all variation in the arabidopsis species. I think of a genome being a thing that each individual has, and that the species can be thought of as having a meta-genome, or just a bunch of genomic variation, but not a singular "genome".

There are several places where grammar needs polishing, for example:

- line 25: evidences -> evidence

- line 45: that -> those

- line 66: change colon to semicolon

- line 88-95: this reads more like a discussion paragraph, summarizing findings before they have even been presented (i.e. line 94-95).

- Figure 1B: It's unclear how the alleles are represented: it says that x's are one allele and o's are the other, but what are heterozygotes?

- I would appreciate a bit more description of how to interpret the ABBA-BABA and f3 statistics and significance, either in the results or methods. These methods are not standard enough that the statistics will be clear to a wide range of readers. What is the cutoff for significance with these tests?

- line 159: define "high relict introgression" quantitatively here.

- line 302-303: "The other relicts belonged to a group that also contained non-relict accession the Mediterranean region and Sweden": Don't the other non-relicts belong to one of the other 4 groups? Which group is this referring to? This paragraph is generally kind of confusing.

Reviewer #2 (Remarks to the Author)

This manuscript from Lee et al is a beautifully pitched and presented contribution to our understanding of an interesting demographic phenomenon in an important, broadly-studied species, *Arabidopsis thaliana* that is rapidly becoming an interesting evolutionary model. The work focuses on demographic turnover and admixture of populations of this human commensal as (most likely) human-associated populations moved through more natural 'relict' populations in the Iberian Peninsula. The work is pitched strongly as a strong example reminiscent of early important demographic turnover events in humans and for this reason, as well as the fact that *thaliana* is such a broadly studied model, will surely be of broad interest.

This is an important, highly original contribution and overall I have few concerns about it as presented. Overall, the data and methodology are very sound and some very creative and interesting novel analysis approaches are employed, especially the assignment of outlier haplotypes to unknown groups (ca ln 222). The conclusions are very robust and tested by several independent methods. References to earlier work are appropriate and judiciously used.

Major point:

The use of QQ plots Figure 2 and S3D should be better explained for those unfamiliar, at least in the methods. It seems surprising that more commonplace enrichment testing methods are not reported for this showcased category (flowering time) and instead an approach that seems relatively obscure to this reader is provided. Can the authors better explain this? In particular, in figure 2, the bend downward in the grey significance threshold may seem very strange to the nonspecialist. Why is flowering time not showing up in Table 1? Whatever the relative powers of these two enrichment methods are, it would help the reader to understand what looks like a discrepancy. If in fact flowering time is not enriched by the analysis in Table 1, I would see this as not detracting from the important main points of the MS; it would simply help for the nonspecialist reader to understand this piece of the story.

Minor points:

Ln 25 'evidences' to 'evidence'

Ln 28 'of species' to 'of the species'

Ln 146 'Tibetan' to 'Tibetans'

Ln 323-324 'human' to 'humans' twice

Ln 364 'therefor' to 'therefore'

Response to reviewers

Reviewer #1 (Remarks to the Author):

The authors have presented a study of patterns of genomic variation and introgression among relict and non-relict populations of *Arabidopsis thaliana*. The subject matter and findings are interesting, and the authors have used some creative methods to study these patterns. But I find that the interpretation seems to go beyond what the data warrants in several important areas, as I describe below in "Major comments" #1 & #2. I like the study of the large effect locus that is highly differentiated with high LD and the description of how this region is introgressed, but there are some issues with respect to the broader use of GWAS that made it difficult to interpret the results (#5, below). The supporting evidence from the chromosomal transposition is quite interesting and it's nice to see that it supports some of the other patterns of introgression. In light of these comments, I think the manuscript requires extensive re-framing, but could be suitable for Nature Communications following such revision. It seems that many of the findings, however, are mainly confirming and extending previous results, so the novelty may not be significant enough for this journal (e.g. references 23-25 previously studied the major effect locus; relict vs. non-relict populations and human-associated spread studied in references 10, 19, 26; identification of the transposition in references 18,29).

Thanks for this positive and constructive review. We respond to the specific comments below. Regarding novelty, the main point of our study is to investigate the detailed population history after non-relict expansion across Eurasia, and we bring in relevant observations (some of which have indeed been published) as needed. For example, refs 23-25 contain GWAS of flowering time, while we provide an evolutionary explanation for the origin of a major polymorphism identified. Similarly, we use the spatial distribution of a previously identified transposition (18, 29) to shed light on history. The observations from human (19, 26) are only included to frame our findings, and have nothing to do with this study. Ref 10, finally, did look at migration history, but with very limited data, and without knowledge of the existence of relict populations.

Major comments:

1. The interpretation seems too ready to assume selection favouring the outlier regions from the relicts without exploring alternative possibilities. For example, presumably, recombination rate affects the degree of introgression or at least the blockiness of introgression, and there are hotspots and coldspots in the genome. It therefore seems possible that the GO term analysis identified genes and GO-terms that co-vary with cold spots or hot-spots? Jumping to adaptive explanations without considering alternatives seems premature. Similarly, I think the outlier-based trends (GWAS + introgression) could plausibly be explained by pure demography

rather than introgression + selection, and more care needs to be taken to recognize this possibility (see #5, below).

Certainly alternative explanations should be considered, but only if they are plausible. As far as we are aware, there is no evidence that recombination is correlated with GO, and also no reason to expect it to be (in this or any other organism). There is evidence that some R genes are recombinational hotspots (Choi et al. 2016), but this is not a general pattern. We are well aware of the problems inherent in these analyses, which is why we base all conclusions on intersection between results (that are as independent as possible in population genetics). For example, the experimental GWAS results (which were of course corrected for population structure; see point 5 below) overlap with the climate associations, as well as with evidence for admixture. Of course these are not strictly independent (the same SNP data are used in all analyses), but the explanation proposed by us seems a lot simpler than any alternative that we are aware of. We have added a note to the Discussion clarifying this.

2a. Assuming the introgression hypothesis is true, what evidence actually pegs the expansion of non-relicts to the original expansion into Europe? There have been many waves of human movement back and forth over the ages, and it seems just as likely that an increase in trade during, say, the middle ages or the industrial revolution, was responsible for the observed introgression patterns (or more likely, very complex patterns of human-associated movement). The authors seem inordinately eager to link this with the Homo sapiens vs. Neanderthal introgression story (e.g. line 170-173) and post-glacial expansion, when there is really only conjecture to support the case one way or the other. I think a lot of reframing and toning down in the Discussion is necessary to be more scientifically rigorous here.

We are certainly not linking the expansion of non-relict *A. thaliana* to the original expansion of anatomically modern humans at the expense of Neanderthals — the latter took place well over 50,000 years ago, much earlier than the non-relict expansion, which is estimated to have occurred during the last 10,000 years (i.e., post-glacially, as in the title). As stated in several places, we are proposing that the non-relict expansion is linked to the spread of agriculture (or, possibly, Indo-europeans), based on timing and the estimated pattern of spread. This is of course mere conjecture, but it is labeled as such.

The Neanderthal introgression story is simply an analogy; a device used to explain the rationale for our study. It is also clearly labeled as such.

2b. In particular, I find the paragraph from lines 336-353 is WAY too speculative. For example, saying that the "northward expansion certainly happened in Sicily and Lebanon relict" is putting too much certainty in these results. This statement arises from the result that $K = 5$ and is based on genotyped accessions, but other relatives that were not included in this panel could have been the source of such northern populations, and if $K \neq 5$ other patterns might be seen. Overall, the paper explores one hypothesis that is consistent with the data, but this does not mean that it is the best explanation, as many other possible hypotheses have not been tested

and much data is not included here. More caveats are necessary with observational studies such as this.

This paragraph and the accompanying figure (Fig. 8) are in the "Discussion" rather than "Results" for a reason: they are meant to summarize our explanation for lot of very complex data and propose the most likely model. However, we agree that the tone is wrong, and we have rewritten it to more clearly label it as *one* model, and make sure readers understand that other models are possible.

In this study, we have clearly considered other hypotheses and tried our best to rule them out. For example, line 213 to 235, and the whole analysis for Figure 4 was designed to answer whether northern outlier haplotypes descended from southern relicts or from unknown northern relicts. The results support the former. Regarding the different K values, one may increase the K value until one finds north is different from south, but in Materials and Methods we have stated why K=5 is a good choice. As we added in the manuscript, K = 6 and 7 have similar pattern as K = 5.

3. While the explanation for the pattern observed in Figure 3C seems consistent with the data (line 186-195), other explanations might also make sense. The fundamental pattern here is that when you identify regions of the genome that are outliers in terms of genetic distance, then count the number of outliers, you see more outliers in extreme northern and southern regions. As noted on line 200, the same pattern could be seen with isolation-by-distance, but the authors discount this because the geographic distance is longer on the east-west axis than the north-south axis. I do not find this discounting entirely convincing, because presumably as a human associated plant, it has been dispersing with human migration, which may have gone much more often east-west than north-south. Geographic distance is less appropriate than "human migration distance" (or something like that). As I mentioned above, human migration is complicated and the simplest story may not be the most accurate one. I find that there is a bit of "just-so story" here and lack of testing against other hypotheses, and I would like to see much more hedging.

We feel that we have tried to rule out the most obvious alternative hypotheses. For example, while it is correct that simple north-south isolation by distance can cause the pattern that more outlier haplotypes are observed in the northern and southern end of species range, such a model would also lead to major population structure separating north from south, rather than east from west as observed by all studies. Furthermore, under this north-south isolation-by-distance scenario one would expect outlier haplotypes in the north to be very genetically distant from southern outlier haplotypes. But as our further analyses showed (Figure 4), a large proportion of northern outlier haplotypes are similar to those identified in the south. These results do not support the north-south isolation-by-distance scenario. We have added this to the manuscript.

On general level, we agree that you can never be certain about these historical models, and we have made changes throughout to clarify this. However, we think we have considerable

evidence for a very interesting model that will likely guide further research, and that is the purpose of models.

4. I am not a fan of the “methods last” style of journal article that has become more popular in the past 10 or so years, but this is hardly the authors' fault. However, I think it would be helpful to have at least one paragraph summarizing the approaches that are being used, perhaps at the end of the introduction. The results section feels like a very abrupt transition. In some cases there are results that are introduced with little context for how the method work. For for example, what are "outlier haplotypes" (line 179)? I can guess, but going to the methods should only be necessary to understand the details of the method, not to understand roughly how it is working. Please try to put more context for the methods in the actual paper, as I don't think page counts are very restrictive for this journal.

We agree, and have added descriptions in the appropriate parts of the results.

5. How was the GWAS done? The methods are very sparse. What program was used? Was there a correction for population structure? If not, some consideration must be made for how co-variation between environment and underlying population structure may bias the results. The lack of detail here made it difficult to assess some of the findings. If population structure covarying with environment is driving the GWAS patterns, then presumably this could also be driving many of the patterns of covariation with blocks of extreme introgression. I think this is unlikely to explain the particularly strong single locus effect, but could be much more important for the other results. For example, on lines 150-157, "the top 25% of SNPs in terms of introgression are more likely to be associated with flowering time than are random SNPs". Assuming there has been no correction for population structure in the GWAS, couldn't this be driven by drift? The phenotype varies along a spatial gradient, as does introgression. Therefore, spurious associations between neutral alleles and phenotypes could occur just due to drift, and these would also be likely to be outliers in terms of introgression. GWAS without correction for population structure will just identify those loci that are most strongly structured in such case, so of course the same ones would be found in each analysis. I couldn't see any way that the analysis had accounted for this very important issue.

We used a mixed linear model with a kinship term (Korte et al., *Nature Genetics*, 2012) to correct for population structure in all GWAS. This information has been added. We apologize for the omission: correcting for population structure is absolutely essential, and we always do it. Evidently it was so obvious to us that we neglected to mention it! We believe this correction addresses the remaining comments.

Minor comments:

6. - line 22: the genome is a product of a dynamic equilibrium, but is not itself an

equilibrium. It is a physical thing. - this use of “genome” seems strange, as it represents all variation in the arabidopsis species. I think of a genome being a thing that each individual has, and that the species can be thought of as having a meta-genome, or just a bunch of genomic variation, but not a singular “genome”.

We have rephrased this part.

7. There are several places where grammar needs polishing, for example:

- line 25: evidences -> evidence
- line 45: that -> those
- line 66: change colon to semicolon

All corrected.

8. - line 88-95: this reads more like a discussion paragraph, summarizing findings before they have even been presented (i.e. line 94-95).

Here we mention North American accessions being introduced by humans in historical times (in the last few hundred years). This is the result from other studies, not our conclusion from this paper. We simply mention this to justify why we only use samples from Eurasia. In this paper we deal with pre-historical patterns of dispersal in Eurasia, not recent human assisted migrations to other continents.

9. - Figure 1B: It's unclear how the alleles are represented: it says that x's are one allele and o's are the other, but what are heterozygotes?

Arabidopsis thaliana is highly selfing and there is essentially no heterozygosity in the analyzed individuals.

10. - I would appreciate a bit more description of how to interpret the ABBA-BABA and f_3 statistics and significance, either in the results or methods. These methods are not standard enough that the statistics will be clear to a wide range of readers. What is the cutoff for significance with these tests?

We added some explanation in the Results section while leaving detailed information in Materials and Methods.

11. - line 159: define “high relict introgression” quantitatively here.

We used the same criteria (top 25% windows with highest relict introgression) as the SNP-based analysis in the previous paragraph. We added this information.

12. - line 302-303: "The other relicts belonged to a group that also contained non-relict accession the Mediterranean region and Sweden": Don't the other non-relicts belong to one of the other 4 groups? Which group is this referring to? This paragraph is generally kind of confusing.

We are referring to the orange haplogroup in Figure 6A. We have modified this paragraph to make the context more clear.

Reviewer #2 (Remarks to the Author):

This manuscript from Lee et al is a beautifully pitched and presented contribution to our understanding of an interesting demographic phenomenon in an important, broadly-studied species, *Arabidopsis thaliana* that is rapidly becoming an interesting evolutionary model. The work focuses on demographic turnover and admixture of populations of this human commensal as (most likely) human-associated populations moved through more natural 'relict' populations in the Iberian Peninsula. The work is pitched strongly as an strong example reminiscent of early important demographic turnover events in humans and for this reason, as well as the fact that *thaliana* is such a broadly studied model, will surely be of broad interest.

This is an important, highly original contribution and overall I have few concerns about it as presented. Overall, the data and methodology are very sound and some very creative and interesting novel analysis approaches are employed, especially the assignment of outlier haplotypes to unknown groups (ca ln 222). The conclusions are very robust and tested by several independent methods. References to earlier work are appropriate and judiciously used.

Major point:

The use of QQ plots Figure 2 and S3D should be better explained for those unfamiliar, at least in the methods. It seems surprising that more commonplace enrichment testing methods are not reported for this showcased category (flowering time) and instead an approach that seems relatively obscure to this reader is provided. Can the authors better explain this? In particular, in figure 2, the bend downward in the grey significance threshold may seem very strange to the nonspecialist. Why is flowering time not showing up in Table 1? Whatever the relative powers of these two enrichment methods are, it would help the reader to understand what looks like a discrepancy. If in fact flowering time is not enriched by the analysis in Table 1, I would see this as not detracting from the important main points of the MS; it would simply help for the nonspecialist reader to understand this piece of the story.

QQ plots are a standard way of comparing two distributions. There are of course many ways of testing for difference, but most rely on a single summary (e.g. difference in mean) and furthermore assume independent observations. Because of linkage disequilibrium, these p-values are anything but (they are spatially auto-correlated along the chromosomes). For this reason we use a permutation scheme that maintains the SNP positions, but "rotates" the

annotation vectors (GWAS p-values vs high/low introgression statues). We suspect that these autocorrelations may be responsible for the downturn in significance threshold, but confess that we have not researched this further. Similar deviations are often seen in GWAS QQ-plots. We have added this explanation to the Methods — thanks for pointing it out.

As for why flowering time does show up in Table 1, the simplest explanation is that the category as a whole is not over-represented, just small subset that actually harbors polymorphism involved in flowering time variation. Our analyses used standard GO categories, not *ad hoc* lists of known flowering time genes.

Minor points:

Ln 25 'evidences' to 'evidence'

Ln 28 'of species' to 'of the species'

Ln 146 'Tibetan' to 'Tibetans'

Ln 323-324 'human' to 'humans' twice

Ln 364 'therefor' to 'therefore'

All corrected.

Reviewers' Comments:

Reviewer #1 (Remarks to the Author)

The authors have addressed or responded adequately to all of my major comments and the manuscript therefore appears to be suitable for publication, pending a few minor considerations. I find the results quite interesting, even though I still feel that more caveats should be made regarding the accuracy of demographic inference. The model fits best out of those that were considered, but there are an infinite number of untested models and many arabidopsis plants in the world that haven't been included in this study. For example, the analysis underlying figure 5 assumes a simple expansion scenario to infer the center of origin, but if the expansion were more complex and reticulated, this inference would be inaccurate. While the Discussion is improved and the results are discussed as a plausible scenario, I would like to see at least some more explicit acknowledgement of uncertainty and caveats.

I do still find the analogy to the Neanderthals clunky and unhelpful in some places (e.g. line 185: "what if the Neanderthals were extinct?...aren't they? confusing rhetorical device. I think you mean "what if the Neanderthals left no fossils?"), but this is a subjective preference.

Finally, a minor comment: the colour scale on Figures 5 and 7 should be labelled on the figure, especially because they represent different statistics.

Reviewer #1 (Remarks to the Author):

The authors have addressed or responded adequately to all of my major comments and the manuscript therefore appears to be suitable for publication, pending a few minor considerations. I find the results quite interesting, even though I still feel that more caveats should be made regarding the accuracy of demographic inference. The model fits best out of those that were considered, but there are an infinite number of untested models and many arabidopsis plants in the world that haven't been included in this study. For example, the analysis underlying figure 5 assumes a simple expansion scenario to infer the center of origin, but if the expansion were more complex and reticulated, this inference would be inaccurate. While the Discussion is improved and the results are discussed as a plausible scenario, I would like to see at least some more explicit acknowledgement of uncertainty and caveats.

The entire manuscript has been revised to address the overall concerns raised here. As part of this process, the following paragraph has been added to the Discussion:

We emphasize that, although we have done our best to rule out alternative models, the model space in these types of historical analyses is infinite, and it is therefore impossible to “prove” that the scenario in Figure 8 is the right one. It is a model that fits the current data well (and we welcome attempts to find better ones — in particular as more samples from under-sampled regions become available).

I do still find the analogy to the Neanderthals clunky and unhelpful in some places (e.g. line 185: “what if the Neanderthals were extinct?”...aren't they? confusing rhetorical device. I think you mean “what if the Neanderthals left no fossils?”), but this is a subjective preference.

We agree. This has been taken care of in the rewrite mentioned above.

Finally, a minor comment: the colour scale on Figures 5 and 7 should be labelled on the figure, especially because they represent different statistics.

Labels have been added.